# Evaluating water quality impacts on visitation to coastal recreation areas using data derived from cell phone locations

Ryan P. Furey [1,2]*, Nathaniel H. Merrill [1], Josh P. Sawyer [1,2], Kate K. Mulvaney [1], Marisa J. Mazzotta [1]

1 U.S. Environmental Protection Agency, Office of Research and Development, Center for Environmental Measurement and Modeling, Atlantic Coastal Environmental Sciences Division, Narragansett, Rhode Island, United States of America, 2 Oak Ridge Associated Universities (ORAU), Oak Ridge, Tennessee, United States of America

* furey.ryan@epa.gov

**Data Availability Statement:** All files are available from: https://github.com/USEPA/Recreation_Benefits.

## Abstract

Linking human behavior to environmental quality is critical for effective natural resource management. While it is commonly assumed that environmental conditions partially explain variation in visitation to coastal recreation areas across space and time, scarce and inconsistent visitation observations challenge our ability to reveal these connections. With the ubiquity of mobile phone usage, novel sources of digitally derived data are increasingly available at a massive scale. Applications of mobile phone locational data have been effective in research on urban-centric human mobility and transportation, but little work has been conducted on understanding behavioral patterns surrounding dynamic natural resources. We present an application of cell phone locational data to estimate the effects of beach closures, based on measured bacteria threshold exceedances, on visitation to coastal access points. Our results indicate that beach closures on Cape Cod, MA, USA have a significant negative effect on visitation at those beaches with closures, while closures at a sample of coastal access points elsewhere in New England have no detected impact on visitation. Our findings represent geographic mobility patterns for over 7 million unique coastal visits and suggest that closures resulted in approximately 1,800 (0.026%) displaced visits for Cape Cod during the summer season of 2017. We demonstrate the potential for human-mobility data derived from mobile phones to reveal the scale of use and behavior in response to changes in dynamic natural resources. Potential future applications of passively collected geocoded data to human-environmental systems are vast.

## Introduction

Environmental degradation is increasingly recognized as having harmful social and economic consequences. Capturing the mode and magnitude of these consequences relies on measurement of human behavior, but the methods of measurement have constrained the ability to reveal the impacts of environmental quality [1, 2]. Environmental characteristics like aesthetics

**Funding:** The author(s) received no specific funding for this work.

**Competing interests:** The authors have declared that no competing interests exist.

and weather are commonly associated with influencing the demand for coastal recreation; however, the spatial heterogeneity of water resources combined with the lack of visitor counts for coastal access points makes the quantification of visitor behavior especially challenging for this setting [3, 4].

Degraded coastal water quality is a pervasive issue and an important factor in the availability and quality of coastal visits. To protect water quality for swimming purposes, coastal recreation waters in the United States are monitored seasonally for bacterial contamination and subsequently closed to swimming when levels surpass established thresholds. It has been demonstrated that closures can lead to economic and social losses for coastal communities [5–8] and that coastal recreation is sensitive to physical characteristics and changes in climactic conditions [3, 9]. However, few studies have empirically demonstrated the impacts of environmental quality on recreational activity at high spatiotemporal resolution across an entire region. This is largely because of the lack of large-scale, consistent, and ongoing visitation data collections. While effective, visual counts and surveys can be time-consuming and expensive, leading to work that is necessarily constrained across space and time and limited in reproducibility [10–12].

Passively collected geocoded data derived from cell phones provide the digital footprints of human activity. Using this form of data to study the spatiotemporal dynamics of human mobility has garnered considerable attention in recent years and is promising as a measurement instrument to assess the distribution of populations in space and time [10, 11, 13–16]. Cell data from a single service provider can quickly accumulate the activities of millions of people, especially in densely populated urban areas [17]. Newer forms of cell data are increasingly robust, as most cell phones contain GPS units that track locational data far more frequently than calls and texts are performed. This means that location data, in the form of latitude and longitude, are generated each time a device interacts with a network, which happens when a device connects to WiFi, GPS, Bluetooth, and mobile applications, in addition to calls or texts. Given the near universal adoption of mobile phones, cell data presents a compelling data source for investigating and understanding human mobility at a global scale, with much potential for understanding human interaction with natural resources [14, 18–20].

Cell data has been extensively used to analyze transportation infrastructure and commuting patterns [10, 11], human mobility [14, 16, 21–24], transportation mode inference [25], and tourism dynamics and human behavior during special events [26–28]. Only a few studies have used cell data to understand human interaction with natural resources. Yu et al. [29] and Nyhan et al. [30–32] conducted some of the few examples of this type of study, using cell data to estimate exposure to ambient air pollution in urban areas. AirSage, in partnership with the U.S. Forest Service and National Park Service, piloted a project to assess monthly visitation to national forests and national parks using AirSage's proprietary data [33]. Merrill et al. [13], Kubo et al. [34], and Monz et al. [35] provided the first uses of cell data to quantify visitation to natural areas across entire regions and for extensive timeframes. While these projects demonstrate the growing use of cell data for understanding general human mobility trends, there has yet to be research which employs cell data to analyze variations in human mobility patterns in response to changes in the quality of coastal resources. To date, no research has used cell data to investigate behavioral responses to water quality changes.

This paper presents an application of human-mobility data derived from mobile phone locations (hereafter referred to as cell data) to estimate the effects of beach closures bacteria exceedances on visitation to 565 coastal recreation areas in New England, USA, 465 of which are on Cape Cod, MA. Merrill et al. [13] demonstrated the viability of cell data to provide extensive and detailed visitation data to natural areas, showing how the data can replicate visitation estimates produced by observational counts, but with a much higher spatial and

temporal resolution and larger geographic extent. Using a dataset derived from cell phone location data which estimates visitation to 565 coastal recreation areas in New England across the summer season of 2017 (June-September), combined with EPA data on beach closures from bacterial contamination, this paper estimates visitation totals and the behavioral impacts of closures for coastal recreation areas on Cape Cod, MA, USA, as well as across New England. Our work demonstrates the potential for cell data to reveal behavioral patterns in response to a dynamic natural resource.

## Materials and methods

### Study area

Cape Cod is a peninsular land mass that protrudes into the Atlantic Ocean from Massachusetts' southeastern shoreline. The coastline is roughly 560 miles long and contains a range of water recreation areas from marine bathing beaches to estuarine waterways and inland ponds. These water recreation areas on Cape Cod are major attractions and provide significant ecosystem services for both visitors and residents alike [36]. The nature of water recreation in New England creates an element of significant seasonality; visitors flood to Cape Cod in the summer months, driving high rates of second-home ownership and residential dependence on a tourism-based economy. In 2015, the Cape Cod Commission estimated that roughly 5 million people visited Cape Cod, more than half of which were sometime between Memorial Day and Labor Day [37]. Given the significance of Cape Cod's water resources to both its seasonal visitors and permanent residents, beach closures from bacterial contamination are a primary concern for local and state environmental managers. While our research is primarily focused on Cape Cod, we include an additional analysis on a smaller sample of coastal recreation areas across the New England states of Maine, New Hampshire, Massachusetts, Rhode Island, and Connecticut. The purpose of this additional analysis was to test how generalizable our results from Cape Cod were across a more diverse range of coastal recreation areas.

### Data

**Cell phone data.**   We purchased cell data to estimate visitation for a comprehensive set of 465 public coastal access points on Cape Cod (Fig 1) during the summer season of 2017 (June, July, August, and September). Our sample of 465 coastal access points on Cape Cod represents all public ways to water which includes monitored and unmonitored freshwater and saltwater recreation areas. These recreation areas vary in type from small inland ponds to large coastal bathing beaches. Monitored recreation areas refer to coastal access points that were routinely (often weekly) examined for bacterial contamination by beach managers, municipalities, or state departments of health. We selected this complete set of public access points to maximize the opportunity for understanding variations in visitation within a socially and economically significant region with water quality concerns and to have comprehensive information for an entire region. Additionally, we purchased data for a set of 100 coastal access points with water quality monitoring histories in the New England states of Maine, New Hampshire, Massachusetts, Rhode Island, and Connecticut. The sample across New England's coastal states are a mix of saltwater beaches and public access points to saltwater areas that vary in size, type, recreational attributes, and water quality histories.

We purchased data from Airsage, Inc. Airsage is one of many companies that sells a range of data products derived from raw cell phone locations, which are collected by cellular service providers and application developers. The specific product provided to EPA by Airsage was developed using GPS locational information captured by smartphone applications. This data is then anonymized, cleaned, and packaged using proprietary methods to transform the raw cell

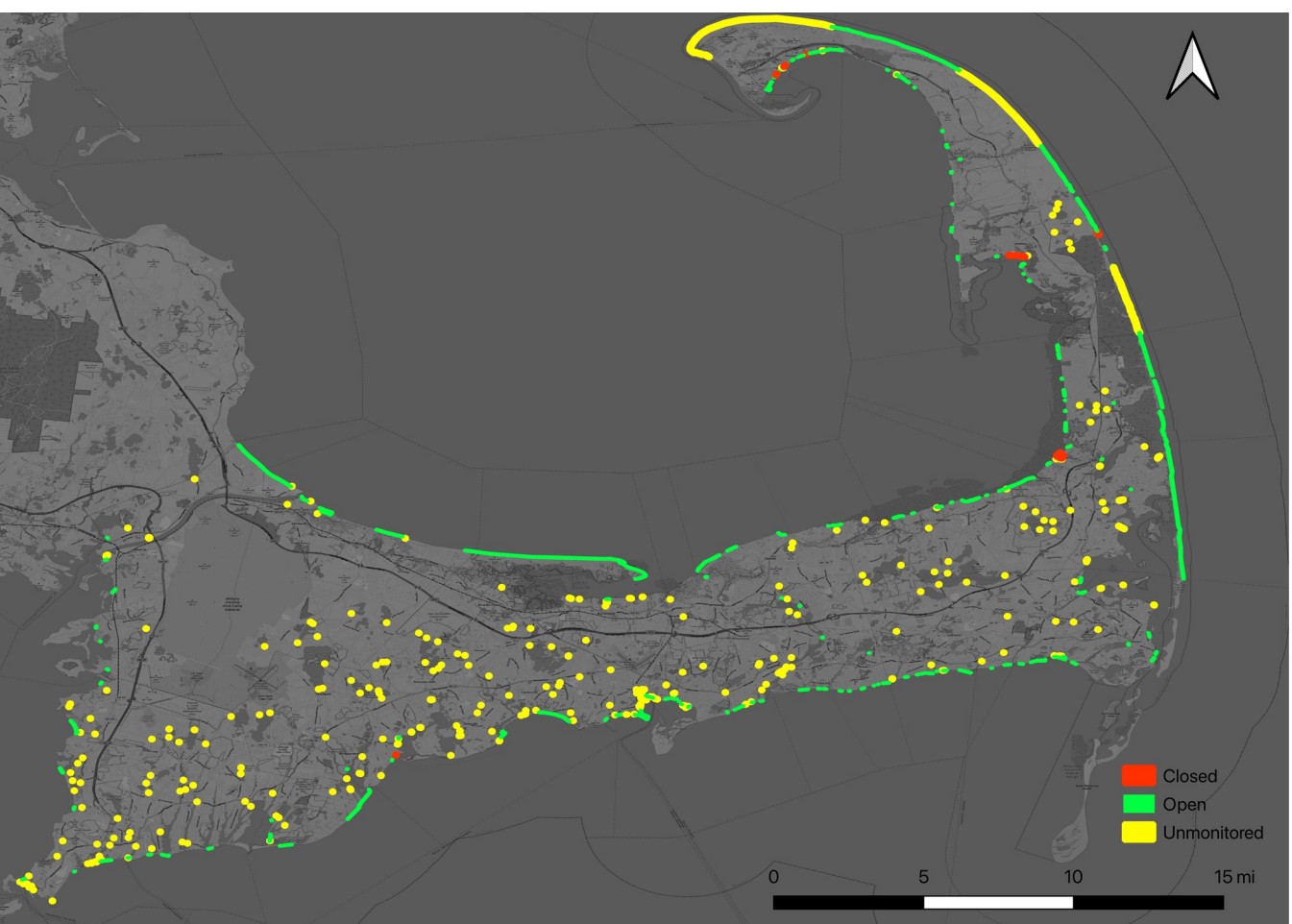

**Fig 1. The comprehensive set of sampled water recreation areas on Cape Cod, Massachusetts.** This includes all public ways to water, from inland freshwater ponds, to estuarine inlets, to large marine swimming beaches. Many of the coastal marine beaches are monitored for bacterial contamination, while freshwater ponds are unmonitored. Closed beaches refer to any beach that was closed, even once, during the summer in 2017. Base map and data from OpenStreetMap and OpenStreetMap Foundation.

data generated by smartphone application-level GPS information into anonymized estimates of visitation, which we aggregated into daily totals.

Through a comparison to a series of observational counts, Merrill et al. [13] determined that the cell data provided by Airsage overestimates the quantity of visitation to the specified geographic areas, especially when aiming to quantify visits that are uniquely recreational. Monz et al. [35] found that calibration of raw cell data was important for park areas in California, as Merrill et al. [13] did for water recreation areas in New England. The Airsage product used in this study was calibrated to observational visitation counts following the process described in Merrill et al. [13] to create daily visitation estimates to all coastal access areas for the duration of the study. Across the entire 2017 summer season, there were 7.5 million distinct visits to the 465 sampled coastal access points on Cape Cod (Fig 2), and 4.3 million visits to the sampled set of 100 coastal access points across New England.

**Beach closure data.** In the United States, non-point source pollution is the most common cause of contaminated water [38]. The specific causes of impairments vary based on location and waterbody type, but most coastal impairment listings are the result of pathogens, specifically bacterial contamination [39]. Bacterial contamination is typically from fecal sources, both

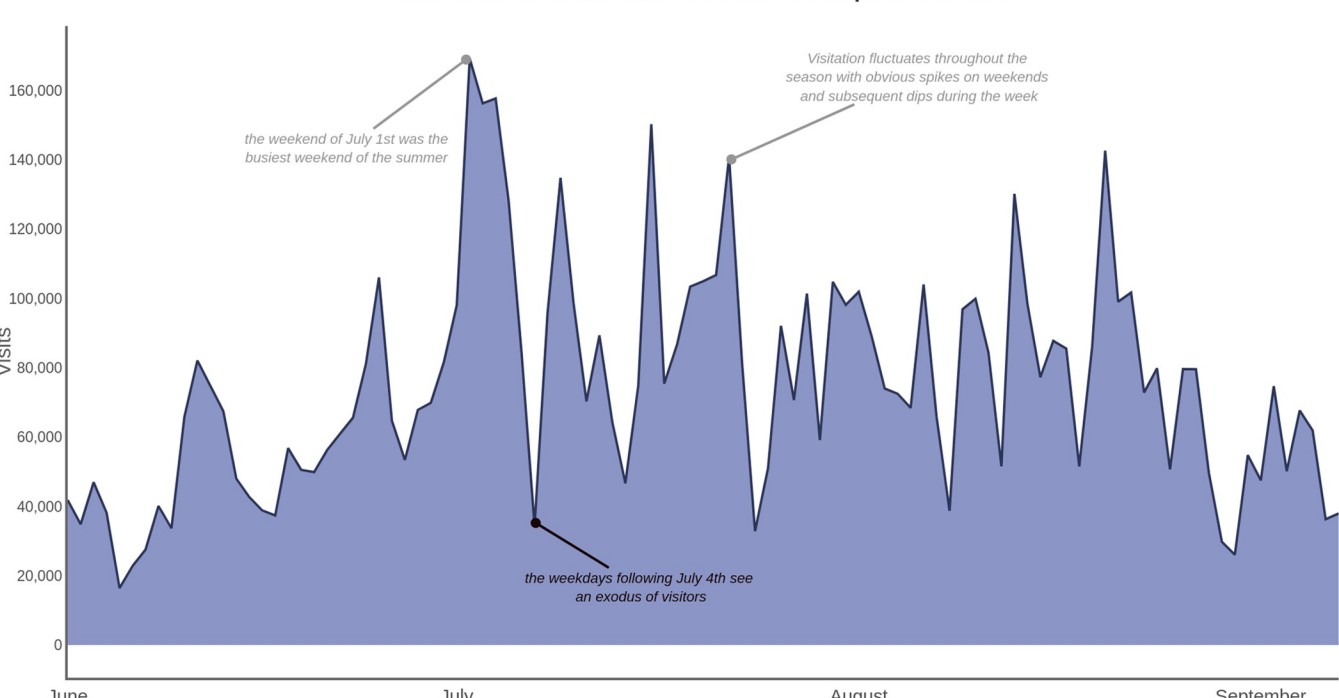

**Fig 2. Total visits to the 465 coastal access points on Cape Cod in 2017.**

point and non-point. The pathways through which these organisms travel are often sewage (leaking sewer pipes, combined sewer overflows, leaching septic systems, etc.), stormwater runoff, or human and animal waste discharged directly into the water. The bacterial pathogens that cause contaminated waters are directly linked to waterborne illnesses that are harmful to humans (such as gastrointestinal illness or respiratory illness) [39]. Given the risk posed to humans who are exposed to these pathogens, federal and state regulations require many coastal recreation areas (mainly popular state saltwater bathing beaches and select freshwater ponds) to be monitored for potential contamination.

The EPA's Beaches Environmental Assessment and Coastal Health (BEACH) Act of 2000 provides states with funding (through BEACH Act grants) to monitor their waterbodies for impairment [40]. The EPA is tasked with providing the data resulting from the states' monitoring efforts to the public. To do this, the EPA developed the Beach Advisory and Closing Online Notification (BEACON) system as a publicly accessible database [41] that aggregates states' beach monitoring data to a national scale. This resource provided the dataset that details 2017 beach closures in New England for our analysis. Beach closure data is readily available and reasonably consistent at a national scale, whereas other water quality measures are not.

On Cape Cod there were 173 water recreation areas monitored for bacterial contamination in 2017. Across the summer season, there were eight closure events at eight (5%) recreation areas resulting in 20 closure days. There were over 800 coastal access points monitored for bacterial contamination across New England in 2017, 251 (nearly 30%) of which were locations that either closed or posted advisories due to water quality testing results surpassing bacterial thresholds. The closures at these 251 locations resulted in 713 closure or advisory days across New England. Further detail on closures across New England is described in the supplementary information.

**Weather data.** Our model also included a single set of weather parameters collected by the National Oceanic and Atmospheric Administration (NOAA) at the Hyannis, Barnstable Municipal-Boardman Airport weather station which is located centrally on Cape Cod and representative of the weather conditions across our sample locations. NOAA provides daily summaries of precipitation, windspeed, and temperature, which we accessed and downloaded through NOAA's online weather data download portal [42].

**Model.** To understand how closures due to impaired water quality affect visits to New England coastal recreation areas, we developed a model that explains the variation in daily visitation to our set of 565 New England coastal access points. Once a representative model was established, we could then interrogate what effect, if any, closures have on visitation to these coastal access points. The closure dataset was incorporated into the behavioral model to determine if closures were significant in explaining the daily variation in visitation across the 2017 summer season conditional on other factors influencing visitation.

The unique size and structure of the visitation data, which includes 51,511 daily visitation estimates, allowed us to apply a panel regression model to understand which factors explain coastal visitation. Taking advantage of the panel structure of our data (location of interest x days), we created a fixed effects regression model to estimate daily visitation as a function of a set of explanatory variables and a coastal access point specific constant:

$$Y_{it} = \alpha_i + \boldsymbol{\beta_W W_t} + \boldsymbol{\beta_M M_t} + \boldsymbol{\beta_D D_t} + \boldsymbol{\beta_H H_t} + \beta_C C_{it} + e_{it}$$

where,

$Y_{it}$—visits to coastal access point $i$ on day $t$ derived from cell data

$\alpha_i$—intercept for each coastal access point $i$

$\boldsymbol{\beta_W W_t}$—vector product of coefficients $\boldsymbol{\beta_W}$ and daily weather conditions $\boldsymbol{W_t}$ (temperature, precipitation, and rainy-day dummy variable)

$\boldsymbol{\beta_M M_t}$—vector product coefficients $\boldsymbol{\beta_M}$ and month dummy variables $\boldsymbol{M_t}$

$\boldsymbol{\beta_D D_t}$—vector product of coefficients $\boldsymbol{\beta_D}$ and day of the week dummy variables $\boldsymbol{D_t}$

$\boldsymbol{\beta_H H_t}$—vector product of coefficients $\boldsymbol{\beta_H}$ and dummy variables for weekends (including holiday weekends) $\boldsymbol{H_t}$

$\beta_C C_{it}$—product of coefficient $\beta_C$ and dummy variable for each day $t$ a beach $i$ had a closure posted $C_{it}$

$e_{it}$—within beach error term

A fixed effect specification controls for any non-time-varying attributes of the coastal recreation areas and points of interest, such as site size, facilities and any other non-varying environmental and site quality features [43]. While the specification controls for these factors in estimating the other marginal effects of interest, it did not allow us to distinguish the individual effect of these non-time varying factors on visitation. Our water quality attribute of interest varied over time, as a time series of open or closed statuses for each coastal access point.

We specified different functional forms of the model: linear, log-linear and log-log. We inferred that all the covariates, such as weather or day of the week, would not explain visitation linearly across differently sized or types of locations, meaning the effect of changes in covariates were not additive but more likely multiplicative. Based on this logic and the structure of the errors post estimation, we chose a log-linear regression model, which fit the data best. A percent change in visitation (*Y*) resulting from a change in one of the explanatory variables, holding all others constant, was calculated as $100 \cdot (e^{\beta_i} - 1)$. Therefore, the effect can be interpreted as the percent fewer people visiting the access point that day than would have been visiting with no closure.

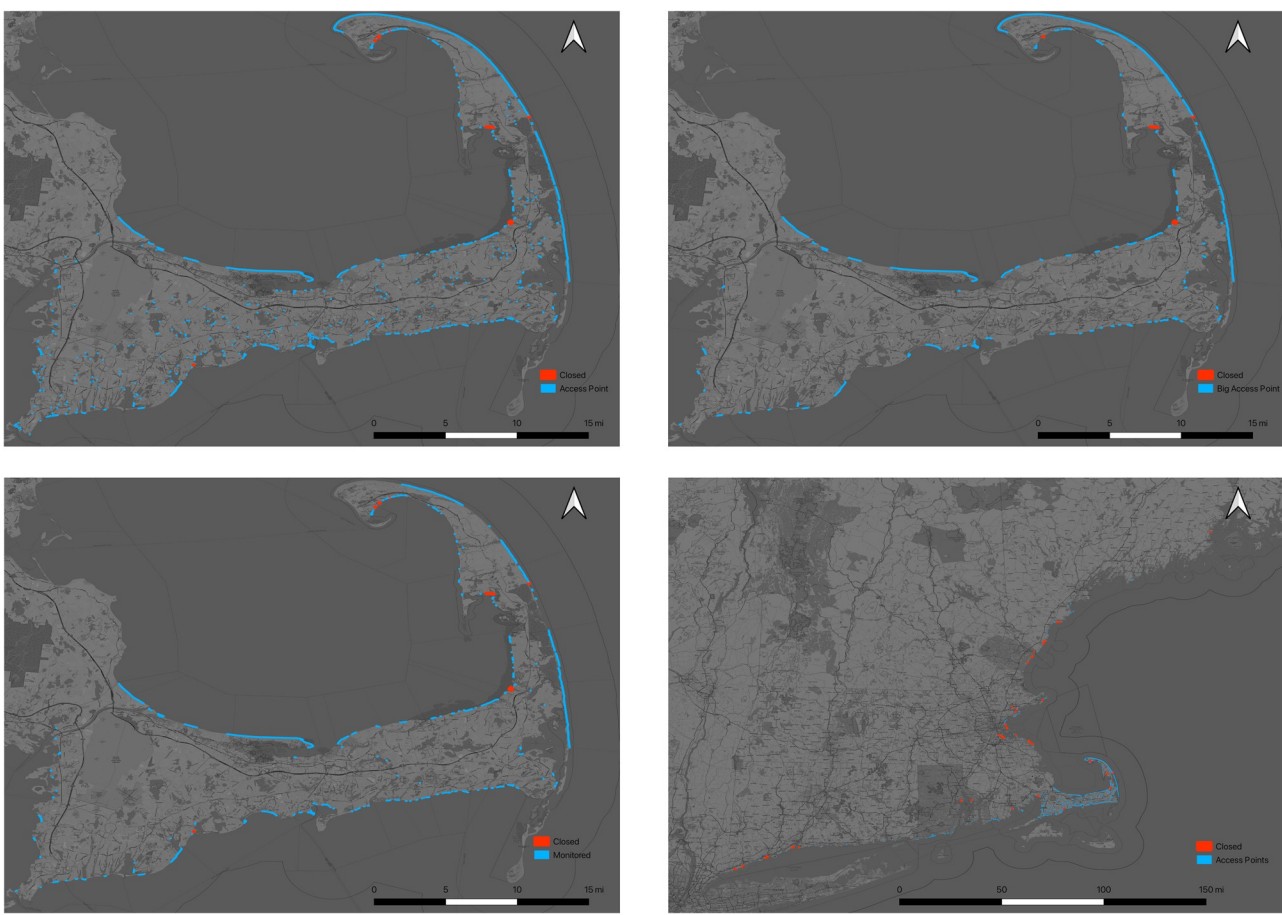

**Fig 3. The four groupings of access points by category.** We ran regressions on each set of coastal access points consisting of (clockwise from top left) all Cape Cod access points ($n$ = 465), all large Cape Cod access points ($3^{rd}$ quartile area m$^2$, $n$ = 185), all off-Cape access points, and all monitored Cape access points ($n$ = 174). Access points with closures in 2017 are indicated in red. Base map and data from OpenStreetMap and OpenStreetMap Foundation.

We ran the regression described above for the set of coastal access points on Cape Cod (n = 465) [45]. In addition, considering the diverse physical and spatial characteristics of the access points in our sample, we elected to stratify the sample into categories (Fig 3) to test two hypotheses: 1. assuming variation in physical and spatial characteristics drives differences in visitor type (and therefore recreational use type), our model would likely explain visitation differently for various categories of water recreation areas; and 2. given the variation in visitation type, we would also expect to capture closures' effect differently for these categories. So, in addition to running the regression on the sample set for Cape Cod, we also ran the regression on spatially large access points on Cape Cod ($3^{rd}$ quartile area m$^2$, n = 185) and on only access points monitored by the Barnstable County Department of Health and Environment for bathing beach quality (bacteria) on Cape Cod (n = 174). Lastly, to investigate how generalizable the results are to access points across New England we also ran the regression on a set of 100 water recreation areas across New England (i.e., off-Cape access points).

## Results

Results from our models can be seen in Table 1 and in the expanded results in the supplementary information. Our initial regression included the entire set of 465 access points on Cape

**Table 1. Fixed effect regressions for beach closures.**

| | Dependent variable: Log of visits | | | |
|---|---|---|---|---|
| | **All Cape Access Points** | **Big Cape Access Points** | **Monitored Cape Access Points** | **Off Cape Access Points** |
| Temperature | 0.080*** | 0.100*** | 0.097*** | 0.090*** |
| | (0.001) | (0.002) | (0.001) | (0.002) |
| Wind | 0.004*** | -0.003 | -0.001 | 0.005* |
| | (0.001) | (0.002) | (0.002) | (0.002) |
| Precipitation | -0.217*** | -0.274*** | -0.112*** | -0.100*** |
| | (0.010) | (0.020) | (0.003) | (0.004) |
| June | -0.217*** | -0.274*** | -0.263*** | -0.167*** |
| | (0.010) | (0.020) | (0.016) | (0.021) |
| July | -0.140*** | -0.145*** | -0.120*** | -0.089*** |
| | (0.011) | (0.023) | (0.019) | (0.024) |
| August | -0.226*** | -0.238*** | -0.230*** | -0.192*** |
| | (0.011) | (0.023) | (0.018) | (0.023) |
| Tuesday | -0.034*** | -0.062*** | -0.068*** | -0.032* |
| | (0.009) | (0.018) | (0.014) | (0.019) |
| Wednesday | 0.074*** | 0.170*** | 0.105*** | 0.116*** |
| | (0.010) | (0.020) | (0.016) | (0.021) |
| Thursday | 0.042*** | 0.028* | -0.003 | 0.018 |
| | (0.009) | (0.017) | (0.014) | (0.018) |
| Friday | 0.095*** | 0.103*** | 0.102*** | 0.131*** |
| | (0.009) | (0.018) | (0.014) | (0.019) |
| Saturday | 0.263*** | 0.273*** | 0.278*** | 0.263*** |
| | (0.010) | (0.020) | (0.016) | (0.020) |
| Sunday | 0.316*** | 0.415*** | 0.386*** | 0.447*** |
| | (0.009) | (0.018) | (0.014) | (0.019) |
| **Closed** | **-0.164**\*\* | **-0.253**\*\* | **-0.180**\*\* | **0.015** |
| | **(0.074)** | **(0.100)** | **(0.083)** | **(0.027)** |
| Observations | 29,141 | 8,565 | 13,360 | 8,196 |
| R² | 0.49 | 0.55 | 0.533 | 0.510 |
| Adjusted R² | 0.478 | 0.538 | 0.526 | 0.503 |
| F Statistic | 1,179.208*** (df = 23; 28701) | 439.217*** (df = 23; 8437) | 653.605*** (df = 23; 13164) | 365.458*** (df = 23; 8073) |

*p<0.1

**p<0.05

***p<0.01.

*Note*: Regressions include fixed effects for each beach. The effect of beach closures is significant for the Cape beaches and more-so for larger beaches over 56,926 m² in size. Each fixed effect regression contains lags of visitation for 10 days to correct for serial correlation in the error term.

Cod across the summer season of 2017. Daily precipitation and temperature had a substantial effect on visitation, with one centimeter of precipitation reducing estimated visitation by 24 percent and an increase in one-degree centigrade resulting in an 8 percent increase. Daily average wind speed (wind), and dummies for months and days of the week were also significant in affecting visitation.

When executing the regression for all Cape Cod access points, spatially large access points on Cape Cod, and monitored access points on Cape Cod, the closure variable was negative and significant (p<0.01). According to our model, a closure at any Cape Cod access point would reduce visitation to that location by 18 percent for that day, while a closure at a large access

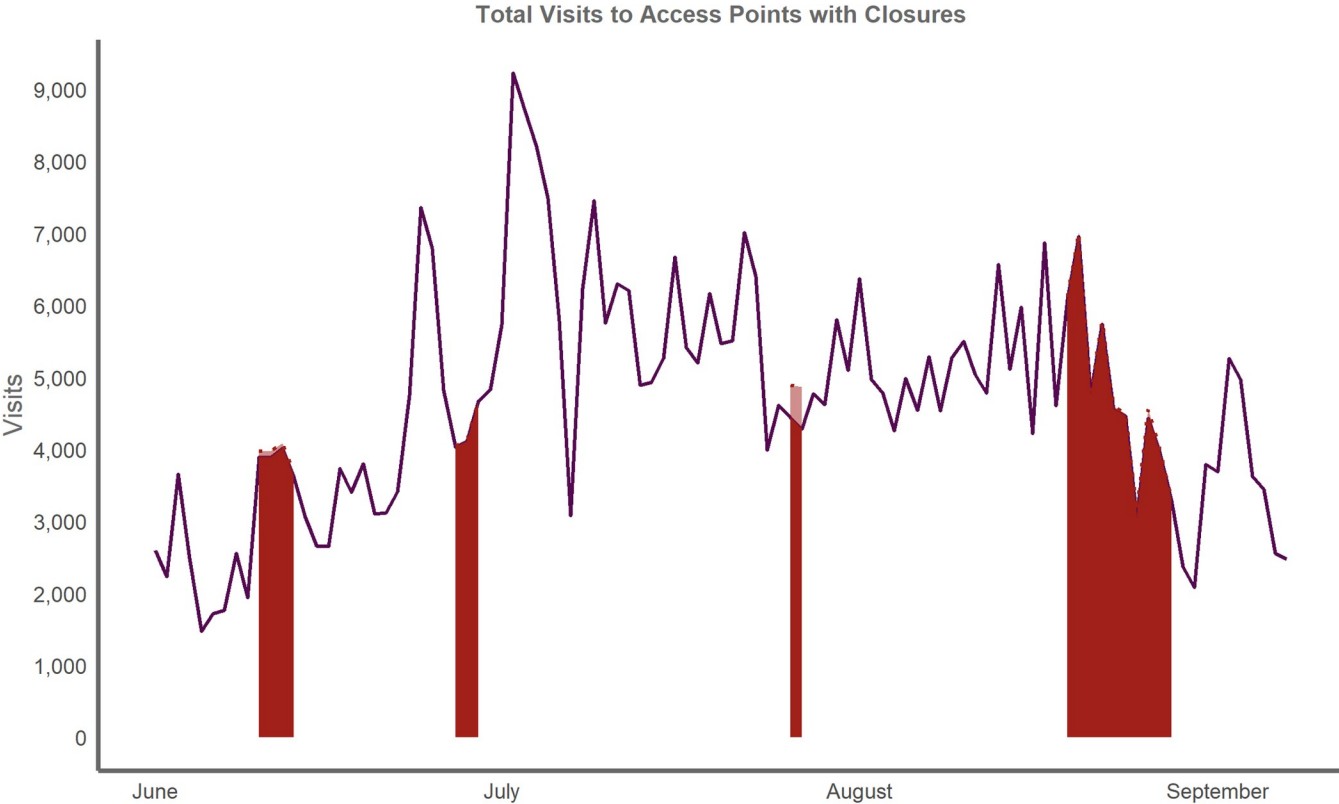

**Fig 4. Total visits to Cape Cod access points with at least 1 closure in 2017.** Red areas beneath the line represent date ranges that have at least one closed beach. We project the total visits if there was not a closure in light red, but the difference is quite small except for the closure in late July.

point on Cape Cod would reduce visitation by 29 percent, and a closure at any monitored access point would reduce visits by 20 percent. In 2017, there were eight coastal access points that had closures from bacterial contamination resulting in 20 days closed to water-based activities. This resulted in approximately 1,800 lost visits for these access points across 20 closure days in the season, shown as the lighter shaded area below the dotted line in Fig 4.

Closure events vary in duration and geographic effect. Looking at individual beaches illuminates the varying effects of closures. Certain events, like the closure at Dyer Street Beach (Fig 5), closed the recreation area for four days, but the closure was isolated to the single beach. Other closure events were minimal in duration, like the event on July 26 and 27, 2017, but affect groups of access points as opposed to isolated locations likely due to regionally high rainfall and runoff events (Fig 6).

Despite the closures being significant and negative for coastal access points on Cape Cod, this result did not generalize to our sample of 100 coastal access points across New England. While there is certainly room to improve upon our model of visitation to more accurately estimate the variation in visitation at each beach, there are several other reasons why effects of closures may not have been detected when running the regression on that set of 100 coastal access points across New England. The beaches where we detected the effect of a closure historically close less frequently (0.4 days per year on average in the last five years for monitored beaches on Cape Cod). Locations that had closures in 2017 where closures were *not* detected as a significant driver of variation in visitation on average close more often (3.3 days per year on average in the last five years for the set of 100 coastal recreation areas across New England). Certain

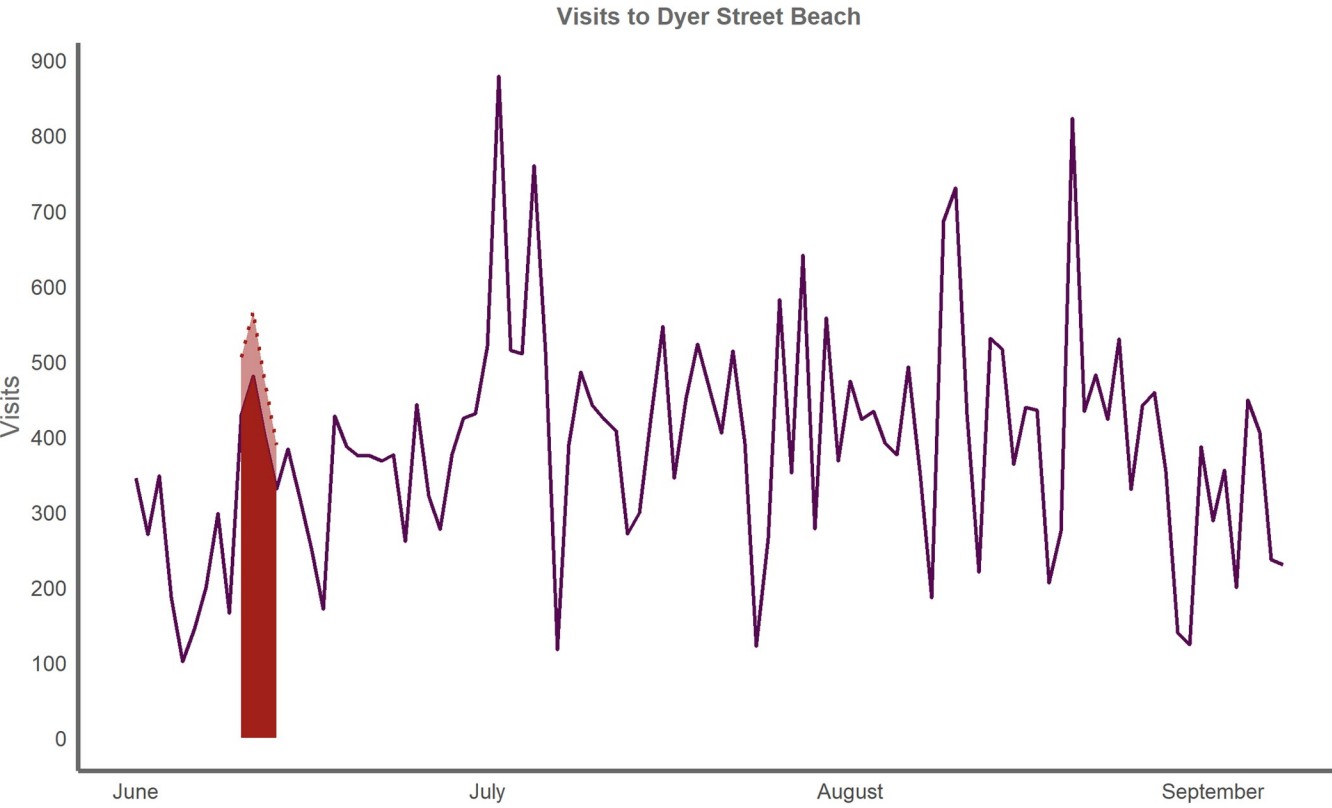

**Fig 5. Visits to Dyer Street beach in 2017.** Red area beneath the line shows the range of dates when the beach was closed. We project the total visits if there was not a closure (i.e., lost visits) in light red below the dotted line.

locations like Wollaston Beach, MA, had five-year closure averages surpassing 30 days annually. We hypothesize that for beaches where closures were detected as significant, the closure was a rarity and resulted in more disruption of assumed quality and water-based activities. For beaches where closures were more frequent, a closure might not have affected the plans of those individuals visiting because the intended activities were not water-based or did not involve direct water contact. Furthermore, those visiting beaches with historically frequent closures may have been aware of that beach's closure reputation and planned their water-contact accordingly. In general, coastal recreation activities on Cape Cod may be more water contact based, where coastal recreation across greater New England may favor activities with little direct water contact. It is difficult to prove this using cell data alone, as there is no straightforward way to stratify visits by activity type. Regardless, these findings point to the limits of using a single general scale of an impact of a beach closure, out of sample, for places where we do not have visitation or recreational behavior information. The use of cell phone locational data allows for vastly more beach-specific visitation data in many more places, limiting the need for applying mean effects from different studies, regions, or beaches.

Another critical driver of our ability to detect the impact of water quality on visitation is the closure dataset itself. While the dataset provides a crucial indicator of water quality at a national scale, the nuances of beach closures vary at the regional, state, and municipal levels. Certain states (like Maine, New Hampshire, and Rhode Island) post "advisories" that are suggestions to beach-goers to avoid contact with water. These states' laws contain provisions where beach postings are the responsibility of local jurisdictions [44, 45]. Other states (such as

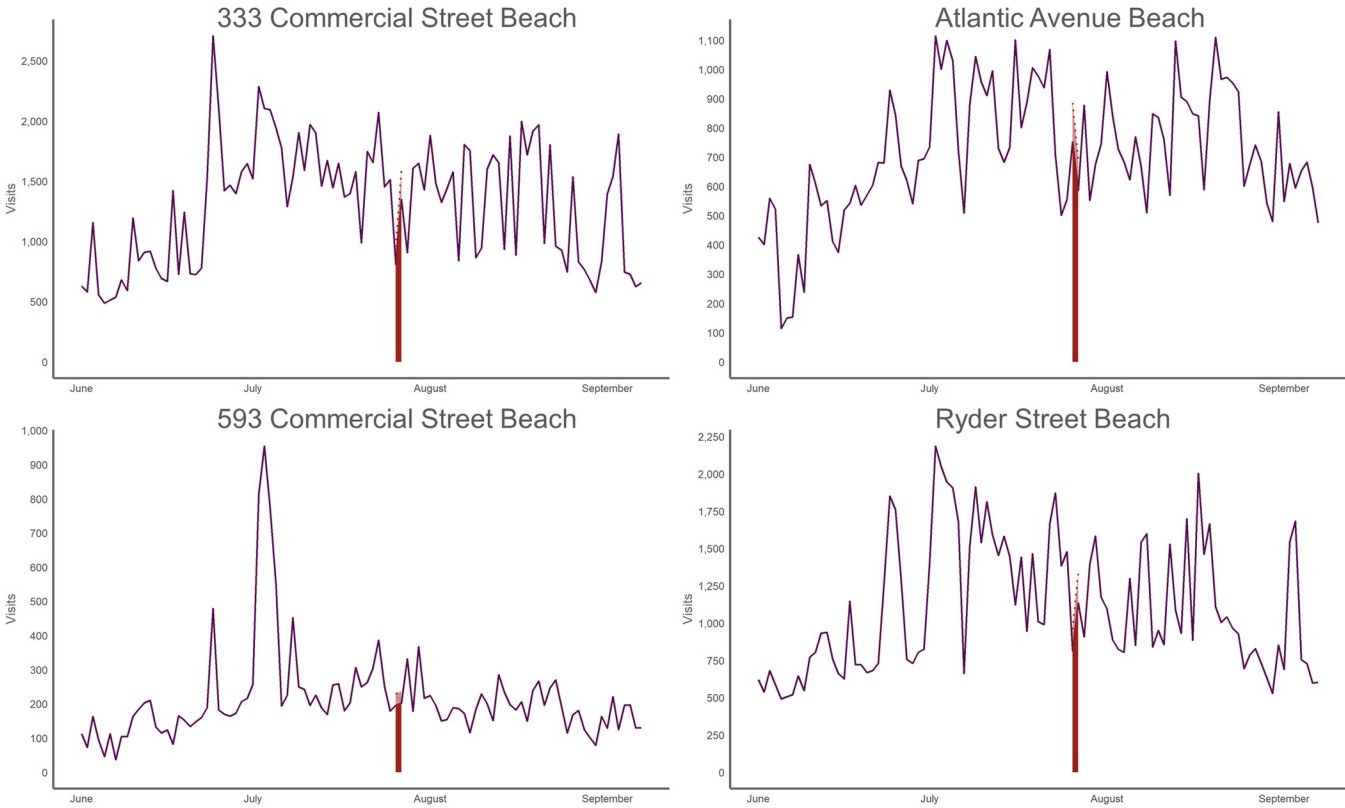

**Fig 6. Seasonal total visits to 333 Commercial Beach, Atlantic Avenue Beach, Ryder Street Beach, and 593 Commercial Street Beach.** Highlighted in red is the July 26 and 27 closure that impacted all four of these beaches. These four simultaneous closure events resulted in the most displaced visits in 2017 of all closure events and were likely directly tied to the same specific rain event.

Massachusetts) have it written into law that swimming or bathing are prohibited when water quality does not meet requirements [46]. In New Hampshire and Maine, local beach managers and boards of health retain the right to keep a beach open or state a beach is closed using their own discretion [45, 47]. Across New England, beach closures and swimming advisories do not prohibit the use of a beach for land-based activities (walking, sports, etc.) [48]. Additionally, the methods of sampling and testing water quality often take at minimum 24 hours to process, creating a potential delay between the beach closure and the quality event.

The aggregation of this ambiguity in the mechanisms employed to measure, close a given beach, and communicate that closure to the public create practical challenges to using beach closures as a proxy for water quality. For example, the impact being measured in this paper is the impact of the whole chain of events from sample to implementation of the policy threshold to communication with the public, through to the closure announcement, signage, and enforcement. The effects being measured in this paper may be distinct from the impact of water quality alone in absence of a closure management system built around it. Other measures of coastal water quality exist (dissolved oxygen, chlorophyll a, Secchi depth, etc.) but they are not collected comprehensively near the recreational areas of interest at a fine time resolution and are not often communicated and shared with the public. These other metrics may also have less of a direct impact on people's day to day decisions around beach water quality than bacteria conditions, given the health implications.

One of the benefits of using more comprehensive datasets of visitation to water recreation areas, as in this study, is quantifying the total use and the number trips protected by town,

regional, and state health departments sampling and closure programs. For Cape Cod, the monitoring program was protective of 4.2 million trips in one summer alone. As coastal recreation continues to grow in its economic influence and cultural value, so should the efficacy of these methods and the funding that is allocated to the organizations responsible for performing them.

## Discussion

This study used cell data to estimate the effects of beach closures from bacteria exceedances on visitation to coastal access points on Cape Cod and in New England, USA for the summer season of 2017. Using a model of daily visitation combined with the dataset on bacterial closures, we were able to detect the impact of closures for coastal access points on Cape Cod. Our findings represent geographic mobility patterns for over 7 million unique visits and suggest that beach closures resulted in approximately 1,800 displaced visits on Cape Cod beaches during the summer season of 2017. However, we were unable to detect this effect for our broader New England sample.

Although there has been significant progress made towards understanding the biological and physical implications of water quality degradation, capturing both market and non-market damages of pollution, especially for water quality degradation, has remained a challenge for researchers. Linking changes in environmental systems to socioeconomic, behavioral, and human health outcomes is a crucial step in valuing the benefits of scientific progress and assessing the damages of environmental degradation [49]. In order to accomplish this, it is necessary to develop methods that reveal the scale of use and behavior in response to changes in dynamic natural resources. As coastal recreation is increasingly recognized as a primary driver of economic activity both across New England and nationally [50], accurate visitation estimates are crucial in enabling novel research methodologies to advance [12].

Cell data offers a promising tool in approaching and resolving spatiotemporal mobility problems in the environmental sciences. Whereas the use of cell data for investigating human interaction with natural resources is still emerging as a body of literature, the last decade has seen substantial growth in volume and velocity of geographically coded data products [19]. With the growing availability of this type of data, many applications complementary to ours are appearing within social sciences research, urban planning, and public health [32, 33, 35]. As these data types become more accessible and accurate, so will the ability of researchers to open new lines of inquiry and derive novel understanding of human-environmental systems.

Despite its promise as an instrument for measuring human behavior around natural resources, cell data is not a panacea. The utility in a spatiotemporally resolved dataset like that provided by mobile devices is in its ability to understand a sample population's behavior. However, it does little to help with understanding the motivations behind decision making. While cell data does well with estimating aggregate daily visitation, this estimate is based on a sample of cell phone users. Thus, we do not know the specific effect of closures on different demographic groups, or the implications of the demographics of the cell data sample on this specific measure of the impact of water quality and beach closures. To understand these nuanced implications, traditional field-based methods of social science are still required. Evaluated independently, cell data lack nuance and context, leading to premature and one-size-fits-all assumptions. Instead, employing methods of research that reveal motivations are more necessary than ever.

## Supporting information

**S1 Appendix. The Clean Water Act, beach monitoring, and what happens when a sample tests positive for contamination.**
(DOCX)

**S1 Table. Monitoring records by state for the 2017 bathing season.**
(DOCX)

**S2 Table. Comprehensive regression results for multiple subsets of the coastal recreation areas.**
(DOCX)

**S1 Fig. Visits to Cahoon Hollow Beach.** The dark red area beneath the visitation line shows the date range when there was a closure event, and projected visitation (i.e. predicted visitation if there had not been a closure event) is shown in light red below the dotted line.
(TIFF)

**S2 Fig. Visits to Cross Street Beach.** The dark red area beneath the visitation line shows the date range when there was a closure event, and projected visitation (i.e. predicted visitation if there had not been a closure event) is shown in light red below the dotted line.
(TIFF)

**S3 Fig.**
(TIFF)

**S4 Fig.**
(TIFF)

**S5 Fig. Visits to Mayo/Indian Neck beach.** The dark red area beneath the visitation line shows the date range when there was a closure event, and projected visitation (i.e. predicted visitation if there had not been a closure event) is shown in light red.
(TIFF)

**S6 Fig. Visits to comprehensive set of recreation areas in New England in 2017.** Our sample consists of 465 recreation areas on Cape Cod, and 100 recreation areas across Connecticut, Rhode Island, Massachusetts (off Cape Cod), New Hampshire, and Maine. This figure shows visitation for all 565 recreation areas across the summer season.
(TIF)

## Acknowledgments

The views expressed in this article are those of the authors and do not necessarily represent the views or policies of the U.S. Environmental Protection Agency. This contribution is identified by the tracking number ORD-038121 of the Atlantic Coastal and Environmental Sciences Division, Center for Environmental Measurement and Modeling, Office of Research and Development, U.S. Environmental Protection Agency. We would like to thank Wei-Lun Tsai, Jessica Daniel, and Hale Thurston for their thoughtful reviews.

## Author Contributions

**Conceptualization:** Ryan P. Furey, Nathaniel H. Merrill, Kate K. Mulvaney, Marisa J. Mazzotta.

**Data curation:** Ryan P. Furey, Nathaniel H. Merrill.

**Formal analysis:** Ryan P. Furey.

**Investigation:** Ryan P. Furey, Nathaniel H. Merrill.

**Methodology:** Ryan P. Furey, Nathaniel H. Merrill.

**Project administration:** Nathaniel H. Merrill.

**Supervision:** Nathaniel H. Merrill, Kate K. Mulvaney, Marisa J. Mazzotta.

**Validation:** Ryan P. Furey, Nathaniel H. Merrill.

**Visualization:** Ryan P. Furey, Josh P. Sawyer.

**Writing – original draft:** Ryan P. Furey, Nathaniel H. Merrill, Kate K. Mulvaney, Marisa J. Mazzotta.

**Writing – review & editing:** Ryan P. Furey, Nathaniel H. Merrill, Josh P. Sawyer, Kate K. Mulvaney, Marisa J. Mazzotta.

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
