## [Decision Letter · Decision Letter 0]

12 Mar 2021

PONE-D-21-02353

A novel approach to evaluating water quality impacts on visitation to coastal recreation areas on Cape Cod using data derived from cell phone locations.

PLOS ONE

Dear Dr. Furey,

Thank you for submitting your manuscript to PLOS ONE. After careful consideration, we feel that it has merit but does not fully meet PLOS ONE’s publication criteria as it currently stands. Therefore, we invite you to submit a revised version of the manuscript that addresses the points raised during the review process.

We look forward to receiving your revised manuscript.

Kind regards,

Bijeesh Kozhikkodan Veettil

Academic Editor

PLOS ONE

Journal Requirements:

3. We note that Figures 1 and 3 in your submission contain satelliteimages which may be copyrighted. All PLOS content is published under the Creative Commons Attribution License (CC BY 4.0), which means that the manuscript, images, and Supporting Information files will be freely available online, and any third party is permitted to access, download, copy, distribute, and use these materials in any way, even commercially, with proper attribution. For these reasons, we cannot publish previously copyrighted maps or satellite images created using proprietary data, such as Google software (Google Maps, Street View, and Earth). For more information, see our copyright guidelines: http://journals.plos.org/plosone/s/licenses-and-copyright.

(1) You may seek permission from the original copyright holder of Figures 1 and 3 to publish the content specifically under the CC BY 4.0 license. 

Additional Editor Comments:

Our expert reviewers suggested major revisions.

Reviewers' comments:

Reviewer's Responses to Questions

**Comments to the Author**

1. Is the manuscript technically sound, and do the data support the conclusions?

Reviewer #1: Partly

Reviewer #2: Partly

2. Has the statistical analysis been performed appropriately and rigorously? 

Reviewer #1: Yes

Reviewer #2: Yes

3. Have the authors made all data underlying the findings in their manuscript fully available?

Reviewer #1: Yes

Reviewer #2: Yes

4. Is the manuscript presented in an intelligible fashion and written in standard English?

Reviewer #1: Yes

Reviewer #2: Yes

5. Review Comments to the Author

Reviewer #1: The authors use a basic fixed-effects panel regression technique to determine the effect of beach closures (a proxy for water quality) on beach visitation in Cape Cod, MA. They find that beach closures in Cape Cod negatively influence visitation rates, but results are insignificant when the sample is extended to the greater New England region. As the second paper to use cell phone data to assess water recreation visitation rates (Merrill et al., 2020), this paper expands the use of novel cell phone data to a practical application. The authors carefully explain the gaps in the literature and how they aim to fill them. They also clearly describe the study area and statistical approach. Finally, the present their results, discuss concerns readers may have (with a few notable exceptions), and describe ways in which limitations can be improved. This research has broad applications for benefit transfer ecosystem services valuations but should be used with caution.

Although this research is well formulated and the results are presented clearly in this paper, there are a couple of concerns that I believe are worth mentioning. First, I’m concerned about the ability of cell phone data to proxy visitation, as expressed in Merril et al. (2020).

Second, I have concerns about the use of beach closures as a proxy for water quality. Closures aren’t proxying water quality per se, instead they proxy water quality thresholds (i.e. the quality level the EPA has deemed too dangerous for direct use). I believe a longer time series of data is needed to accurately determine the effects of water quality on visitation and here is why:

1. This paper estimates the impact of beach closures due to extreme water quality degradation (above the EPA threshold) on beach visitation, not the impact of water quality on beach visitation. This sounds equivalent but they are not. The EPA threshold influences the results that water quality has on visitation. Visitors may not visit beaches with quality below the threshold because the beach is closed, not because they are avoiding poor water conditions. Change the threshold and the results will likely change. A better way is to include a water quality index and a beach closure dummy variable. i.e. below the water quality threshold., and an interaction. This would allow the authors to determine the impact of closure, i.e. the threshold, on visitation.

2. Beach visitors (especially local/ multiple use day visitors) may use average water quality from past experiences as a proxy for the recreational amenities of each beach. Stated differently, they have rational expectations for beach quality and only large unexpected deviations from mean water quality will influence their choice. One way to test this hypothesis would be to determine where the closure threshold lands on the distribution of water quality for each beach access point and then remove data points where the threshold is, say, 1-2 standard deviations from mean water quality. I suspect access points with poorer average water quality will not see a huge change in visitation rates from closures because visitors already consider the poor water quality while considering other options. I think using the threshold as a proxy underestimates the true impact of water quality on sites where direct water recreation makes up the majority of recreation. Where poor water quality has persisted at a site for a long time, the beach may have already adapted to cater to non-direct water-based recreation so changes in water quality will have little impact on visitation.

Third, the authors may have missed a reason for the difference in the statistical significance of closures as a predictor of visitation in Cape Cod and the greater New England region. The greater New England region (as a whole) probably has relatively little direct water recreation (e.g. swimming and boating) off the coast. I imagine these results would not transfer well to other regions where direct water reactional makes up a large portion of total recreation, which should be stated by the authors.

Finally, demographics likely play a role in the results. Demographics may be influencing the results if cell phone users are more or less likely to avoid beach closures due to poor water quality than non-cell phone users. For example, older individuals may be more reluctant to visit beaches with poor water quality than younger individuals and be less likely to have a cell phone. Thus, the research estimates the effects of beach closures on visitation by cell phone users, not the general public.

In all, I believe this is a well written paper that should be revised to account for the concerns I mentioned in this review.

I am available to view a revised version if necessary.

Reviewer #2: Author proposed a novel approach to evaluate the water quality impacts on visitation to coastal recreation areas. There are some significant improvements required as following:

Line 51-54, author needs to re-write again. It is quite difficult to understand "what the author wants to explain".

Datasets: There are a lot of Datasets used in this study, In line 87-89 authors mentioned about combining these datasets. However, there is a lack of explanation about “how they combine these datasets”. Either author used the graphical explanation or block diagram for the readers to understand easily.

Model: Authors proposed a novel approach is used in the title. But in this study, linear regression model is used or can say modified linear regression model used. If the author used the “novel approach” comparison is required with existing approaches or models.

Mathematical explanation is very weak. It should be improved.

Results: What are technical parameters, parameters for the linear regression model and computer specification for the reproducible of the results ? Results reproducible is very important for the future researchers that follows the same path of results or improves these results outcome.

6. PLOS authors have the option to publish the peer review history of their article (what does this mean?). If published, this will include your full peer review and any attached files.

Reviewer #1: No

Reviewer #2: No

---

## [Author Response · Author response to Decision Letter 0]

22 Dec 2021

Response to Reviewers

Reviewer #1: 

"First, I’m concerned about the ability of cell phone data to proxy visitation, as expressed in Merril et al. (2020)."

We largely share these concerns about the ability of cell phone data to proxy visitation. In our manuscript (line 136), we reference the Merrill et al. (2020) finding that the raw cell phone locational data “out of the package” overestimated visitation to the specified geographic areas. To account for this, Merrill et al. (2020) conducted a series of manual observational counts to calibrate the raw cell data and found that cell data is useful and accurate once rescaled.

We understand there still may be concerns with using cell phone data to proxy visitation. The proprietary process vendors use to take raw signal data and translate it into population-scale aggregates remains opaque. Considering this, we recommend examining raw data from vendors before assuming it is representative. This is one of the reasons why we chose to use a dataset calibrated to on-the-ground observations for the region (Merrill et al. 2020).

While the use of cell phone location data as a measure of recreational visitation to natural areas may still require caution, there are no other sources of visitation estimates that can provide the same spatial scale, temporal resolution, and demonstrated accuracy to pick up a water quality impact, even one such as this, based on a policy threshold.

"Second, I have concerns about the use of beach closures as a proxy for water quality. Closures aren’t proxying water quality per se, instead they proxy water quality thresholds (i.e. the quality level the EPA has deemed too dangerous for direct use). "

These water quality thresholds are implemented specifically to protect the public from exposure to harmful levels of pathogens during water recreation. While it may be a threshold-based measure of water quality (like DO criteria, as another example), it is the best mechanism available at the geographic scale, timeframe, and connection to people that we are interested in examining with the cell phone data. 

Water quality data across an entire region (like Cape Cod, for example, or especially across a larger geographic area that spans municipalities or states like New England) is unavailable at consistent time intervals let alone daily. While it may be ideal to develop a water quality index that combines biochemical data with beach closure data, the biochemical data is currently too incomplete to use for our purposes. In the meantime, bacteria sampling and beach closures represent the closest thing to a daily water quality indicator that we have that is consistent across all of Cape Cod and New England (this sentiment is expressed in our manuscript in line 165) and is directly people relevant.

To address these concerns in the manuscript, we made it more clear the type of water quality metric (based on bacteria conditions and a policy threshold) we are using by changing wording in the introduction. We also added to qualifying the results on page 14 putting the bacteria sampling and closure program in context with other water quality monitoring data options so that the reader can appreciate this point that the impact of closures is due to a whole series of events including, but not limited to, measuring water quality. 

“The aggregation of this ambiguity in the mechanisms employed to measure, close a given beach and communicate that to the public create practical challenges to using beach closures as a proxy for water quality. For example, the impact being measured in this paper is the impact of the whole chain of events from sample to implementation of the policy threshold to communication with the public, through the closure announcement, signage and enforcement. This effect being measured in this paper may be distinct from the impact of water quality alone in absence of a closure management system built around it. Other measures of coastal water quality exist (dissolved oxygen, chlorophyll a, Secchi depth, etc.) but they are not collected comprehensively near the recreational areas of interest at a fine time resolution and are not often communicated and shared with the public. These other metrics may also have less of a direct impact on people’s day to day decisions around beach water quality than bacteria conditions, given the health implications.”

"I believe a longer time series of data is needed to accurately determine the effects of water quality on visitation and here is why:

1. This paper estimates the impact of beach closures due to extreme water quality degradation (above the EPA threshold) on beach visitation, not the impact of water quality on beach visitation. This sounds equivalent but they are not. The EPA threshold influences the results that water quality has on visitation. Visitors may not visit beaches with quality below the threshold because the beach is closed, not because they are avoiding poor water conditions. Change the threshold and the results will likely change. A better way is to include a water quality index and a beach closure dummy variable. i.e. below the water quality threshold., and an interaction. This would allow the authors to determine the impact of closure, i.e. the threshold, on visitation."

It may be the case that changing the threshold would influence the impact of water quality on visitation, since the policy would change. However, developing a daily water quality index for every beach in our sample is not currently possible given dispersed and inconsistent water quality data. Our lab is currently working on this exact problem, but it is an ongoing effort that is outside the scope of this paper. 

We agree that using a water quality index and beach closure dummy variable could be a better approach to disentangle a threshold versus a gradient of conditions So, the results of this paper ask if, and how much, visitation is sensitive to this water quality driven management threshold. We make this clear up front now in the abstract and introduction.

We added text on this important point throughout and in the results on Page 14. Text referenced in the response to the previous comment.

"2. Beach visitors (especially local/ multiple use day visitors) may use average water quality from past experiences as a proxy for the recreational amenities of each beach. Stated differently, they have rational expectations for beach quality and only large unexpected deviations from mean water quality will influence their choice. One way to test this hypothesis would be to determine where the closure threshold lands on the distribution of water quality for each beach access point and then remove data points where the threshold is, say, 1-2 standard deviations from mean water quality. I suspect access points with poorer average water quality will not see a huge change in visitation rates from closures because visitors already consider the poor water quality while considering other options. I think using the threshold as a proxy underestimates the true impact of water quality on sites where direct water recreation makes up the majority of recreation. Where poor water quality has persisted at a site for a long time, the beach may have already adapted to cater to non-direct water-based recreation so changes in water quality will have little impact on visitation."

This is a great idea and something we considered while writing the paper. We make reference to the idea of average closures impacting our ability to detect the effects on visitation in the conclusion (lines 271-285), and similarly discuss the idea of a beach site catering to non-direct water-based recreation based on historic water quality. We hypothesize this average condition and expectations as a potential reason why we did not find the effect of closures with this instrument off the Cape (on relatively dirtier beaches).

"Third, the authors may have missed a reason for the difference in the statistical significance of closures as a predictor of visitation in Cape Cod and the greater New England region. The greater New England region (as a whole) probably has relatively little direct water recreation (e.g. swimming and boating) off the coast. I imagine these results would not transfer well to other regions where direct water reactional makes up a large portion of total recreation, which should be stated by the authors."

We alluded to this hypothesis in the conclusion but have now more explicitly stated this idea in added lines 279-286. To clarify, though, I believe the reviewer may have reversed the implication in the comment here. The reviewer states: “...these results would not transfer well to other regions where direct water recreational makes up a large portion of total recreation,” but I think the inverse may be the case; our results may not transfer well to other regions where direct water recreation makes up a small portion of total recreation, but may transfer well for other regions similar to Cape Cod where water-contact is the dominant activity type. 

We edited and added this text to page 12 and 13:

“Despite the closures being significant and negative for coastal access points on Cape Cod, this result did not generalize to our sample of 100 coastal access points across New England. While there is certainly room to improve upon our model of visitation to more accurately estimate the variation in visitation at each beach, there are several other reasons why effects of closures may not have been detected when running the regression on that set of 100 coastal access points across New England. The beaches where we detected the effect of a closure historically close less frequently (0.4 days per year on average in the last five years for monitored beaches on Cape Cod). Locations that had closures in 2017 where closures were not detected as a significant driver of variation in visitation on average close more often (3.3 days per year on average in the last five years for the set of 100 coastal recreation areas across New England). Certain locations like Wollaston Beach, MA, had five-year closure averages surpassing 30 days annually. We hypothesize that for beaches where closures were detected as significant, the closure was a rarity and resulted in more disruption of assumed quality and water-based activities. For beaches where closures were more frequent, a closure might not have affected the plans of those individuals visiting because the intended activities were not water-based or did not involve direct water contact. Furthermore, those visiting beaches with historically frequent closures may have been aware of that beach’s closure reputation and planned their water-contact accordingly. In general, coastal recreation activities on Cape Cod may be more water contact based, where coastal recreation across greater New England may favor activities with little direct water contact. It is difficult to prove this using cell data alone, as there is no straightforward way to stratify visits by activity type. Regardless, these findings point to the limits of using a single general scale of an impact of a beach closure for places where we do not have visitation or recreational behavior information. The use of cell phone locational data allows for vastly more beach-specific visitation estimates in many more places, limiting the need for applying mean effects from different studies, regions or beaches.”

"Finally, demographics likely play a role in the results. Demographics may be influencing the results if cell phone users are more or less likely to avoid beach closures due to poor water quality than non-cell phone users. For example, older individuals may be more reluctant to visit beaches with poor water quality than younger individuals and be less likely to have a cell phone. Thus, the research estimates the effects of beach closures on visitation by cell phone users, not the general public."

The manual counts observed by Merrill et al. (2020) were demographic agnostic and, therefore, our calibrated visitation counts should be corrected for any bias towards cell phone users in terms of estimating daily visitation. However, if the effect of the closure were to change this relationship between the cell phone users in the sample and visitation (say if demographics affected this impact) than this could be an issue. 

We acknowledge this point now in the discussion on page 16:

“Despite its promise as an instrument for measuring human behavior around natural resources, cell data is not a panacea. The utility in a spatiotemporally resolved dataset like that provided by mobile devices is in its ability to understand a sample population’s behavior. However, it does little to help with understanding the motivations behind decision making. While cell data does well with estimating aggregate daily visitation, this estimate is based on a sample of cell phone users. Thus, we do not know the specific effect of closures on different demographic groups, or the implications of the demographics of the cell data sample on this specific measure of the impact of water quality and beach closures. To understand these nuanced implications, traditional field-based methods of social science are still required. Evaluated independently, cell data lack nuance and context, leading to premature and one-size-fits-all assumptions. Instead, employing methods of research that reveal motivations are more necessary than ever.” 

Reviewer #2: 

Line 51-54, author needs to re-write again. It is quite difficult to understand "what the author wants to explain".

Rewritten to (now in lines 52-54): “However, few studies have empirically demonstrated the impacts from changes in environmental quality on recreational activity at high spatiotemporal resolution across an entire region.”

Datasets: There are a lot of Datasets used in this study, In line 87-89 authors mentioned about combining these datasets. However, there is a lack of explanation about “how they combine these datasets”. Either author used the graphical explanation or block diagram for the readers to understand easily.

There are multiple data sources for this study (beach closures, weather data, cell-phone data), but the model relies on a single dataset which unites by a “date” variable. This dataset is publicly available in a Github repository (see below for link). 

Model: Authors proposed a novel approach is used in the title. But in this study, linear regression model is used or can say modified linear regression model used. If the author used the “novel approach” comparison is required with existing approaches or models.

The novel part is combining cell phone data to estimate the effects of environmental condition. The linear regression is a common tool, but the data inputs and application are novel to the field. The literature review compares this to other applications to park and tourism management. 

We changed the titled to “Evaluating water quality impacts on visitation to coastal recreation areas using data derived from cell phone locations”. We explain the novel contribution of the paper in the introduction and literature review. 

Mathematical explanation is very weak. It should be improved. 

As suggested above, linear regression is standard in many fields, so we kept the explanation to the inputs and specification of the model and not the mathematics of the regression in linear and log forms. The references provided explain the regression model specifics. We also include a code package for more model specifics and reproducibility. 

Results: What are technical parameters, parameters for the linear regression model and computer specification for the reproducible of the results ? Results reproducible is very important for the future researchers that follows the same path of results or improves these results outcome.

We have built a Github repository (available at https://github.com/USEPA/Recreation_Benefits) that includes all technical parameters, computer specifications, and data used in this study.

---

## [Decision Letter · Decision Letter 1]

25 Jan 2022

Evaluating water quality impacts on visitation to coastal recreation areas using data derived from cell phone locations.

PONE-D-21-02353R1

Dear Dr. Furey,

We’re pleased to inform you that your manuscript has been judged scientifically suitable for publication and will be formally accepted for publication once it meets all outstanding technical requirements.

Kind regards,

Bijeesh Kozhikkodan Veettil

Academic Editor

PLOS ONE

Additional Editor Comments (optional):

Reviewers' comments:

Reviewer's Responses to Questions

**Comments to the Author**

1. If the authors have adequately addressed your comments raised in a previous round of review and you feel that this manuscript is now acceptable for publication, you may indicate that here to bypass the “Comments to the Author” section, enter your conflict of interest statement in the “Confidential to Editor” section, and submit your "Accept" recommendation.

Reviewer #2: All comments have been addressed

2. Is the manuscript technically sound, and do the data support the conclusions?

Reviewer #2: Yes

3. Has the statistical analysis been performed appropriately and rigorously? 

Reviewer #2: Yes

4. Have the authors made all data underlying the findings in their manuscript fully available?

Reviewer #2: Yes

5. Is the manuscript presented in an intelligible fashion and written in standard English?

Reviewer #2: Yes

6. Review Comments to the Author

Reviewer #2: (No Response)

7. PLOS authors have the option to publish the peer review history of their article (what does this mean?). If published, this will include your full peer review and any attached files.

Reviewer #2: No

---

## [Editor Report · Acceptance letter]

21 Feb 2022

PONE-D-21-02353R1 

Evaluating water quality impacts on visitation to coastal recreation areas using data derived from cell phone locations. 

Dear Dr. Furey:

I'm pleased to inform you that your manuscript has been deemed suitable for publication in PLOS ONE. Congratulations! Your manuscript is now with our production department. 

Kind regards, 

on behalf of

Dr. Bijeesh Kozhikkodan Veettil 

Academic Editor

PLOS ONE